# Whey Protein Hydrolysates Improved the Oxidative Stability and Water-Holding Capacity of Pork Patties by Reducing Protein Aggregation during Repeated Freeze–Thaw Cycles

**DOI:** 10.3390/foods11142133

**Published:** 2022-07-19

**Authors:** Chunyun Liu, Lingru Kong, Pengjuan Yu, Rongxin Wen, Xiaobo Yu, Xinglian Xu, Xinyan Peng

**Affiliations:** 1College of Life Sciences, Yantai University, Yantai 264005, China; lcyytu2020@163.com (C.L.); guagua2226@163.com (L.K.); yupengjuan99@163.com (P.Y.); 18800434580@163.com (R.W.); 2Key Laboratory of Meat Processing and Quality Control, Ministry of Education, College of Food Science and Technology, Nanjing Agricultural University, Nanjing 210095, China; yuxiaobo@njau.edu.cn (X.Y.); xlxu@njau.edu.cn (X.X.)

**Keywords:** whey protein hydrolysates, pork patties, freeze–thaw cycles, myofibrillar protein, antioxidant activity, water-holding capacity

## Abstract

The effects of whey protein hydrolysates (WPH) on myofibrillar protein (MP) oxidative stability and the aggregation behavior and the water-holding capacity of pork patties during freeze–thaw (F–T) cycles were investigated. During F–T cycles, the total sulfhydryl content and zeta potential of MP decreased, while peroxide value, surface hydrophobicity, particle size, pressure loss and transverse relaxation times increase. The oxidative stability and the water-holding capacity of pork patties were enhanced by the addition of WPH in a dose-dependent manner, whereas the MP aggregation decreased. The addition of 15% WPH had the most obvious effects on the pork patties, which was similar to that of the 0.02% BHA. After nine F–T cycles, the POV, surface hydrophobicity, particle size and pressure loss of the pork patties with 15% WPH were reduced by 17.20%, 30.56%, 34.67% and 13.96%, respectively, while total sulfhydryl content and absolute value of zeta potential increased by 69.62% and 146.14%, respectively. The results showed that adding 15% WPH to pork patties can be an effective method to inhibit lipid and protein oxidation, reducing protein aggregation and improving the water-holding capacity of pork patties during F–T cycles.

## 1. Introduction

Freezing is the most widely accepted and safest method for preservation and storage of meat and meat products [1]. Refrigeration at low-temperature has long been known to prevent the growth of microorganisms, spoilage and biochemical degradation of meat products [2]. Freezing can extend the storage time of meat and meat products, but it cannot inhibit oxidation completely. In the real storage, production and sales process, the lack of a cold chain will cause the frozen meat to inevitably undergo freeze–thaw (F–T) cycles [3,4]. The F–T cycles have an impact on the structure, functionality and quality of meat products, causing water loss and texture and colour deterioration [5]. The food quality deterioration caused by F–T cycles has drawn the attention of researchers in the meat industry. Especially, the demand for chopped pork-based meat products, such as pork patties, meatballs, sausage, hamburgers and fillers, has gradually increased, and the quality deterioration of these products caused by F–T cycles has become a key problem to solve [6,7]. Therefore, it is of great theoretical and practical value to explore the methods to improve the oxidation stability of frozen meat and maintain its quality.

During F–T cycles, protein and lipid oxidation are the important factors responsible for the deterioration of meat quality [8]. Primary lipid oxidation products include mainly hydroperoxides and free radicals. Hydroperoxides easily decompose to secondary oxidation products, such as malondialdehyde, hexanal and furan, which tend to produce an unpleasant flavour [9]. Free radicals usually react with proteins to promote oxidative deterioration of proteins [10]. Myofibrillar protein (MP), as the main constituent of muscle protein and accounting for 55% to 60% of total muscle protein, are essential factors in the quality of meat and meat products [11]. In general, MP aligns in an orderly fashion, forming a compact and uniform three-dimensional gel network after heat treatment [12]. Protein oxidation can increase MP unfolding, leading to protein aggregation, which also alters the intracellular and intercellular structure of MP. On the other hand, protein oxidation can result in free radical formation, exposure of hydrophobic domain, formation of amino acid derivatives and protein polymerization and degradation [13].

MP is primarily responsible for maintaining water in muscle cells [14]. However, protein oxidation reduces the number of water-protein binding sites [15]. The slight damage to the cell membrane caused by protein oxidation may facilitate the movement of internal moisture under the influence of osmotic pressure [16]. Through osmosis, intracellular juice is released into the extracellular region, forming ice crystals. The formation of extracellular ice crystals and the redistribution of water during multiple F–T cycles promote cell rupture and muscle fiber destruction, resulting in significant mechanical damage to the cell membrane and tissue structure [17]. Cheng et al. [18] also confirmed that protein oxidation causes water migration and water loss, resulting in deterioration of texture and sensory perception of meat products, such as tenderness and juiciness. Overall, the extent of meat oxidation denaturation is the most important factor affecting meat quality in the multiple F–T cycles.

In recent years, considering the potential hazards of synthetic antioxidants, natural antioxidant peptides exploited by various proteins have been widely used in frozen meat products to retard lipid and protein oxidation and improve quality [19]. Protein hydrolysates can bind covalently and non-covalently to proteins in meat systems, preventing protein oxidation [20,21,22]. Nikoo et al. [23] demonstrated that proteolytic peptides could efficiently reduce the water loss of MP during F–T cycles. Our previous studies found that whey protein hydrolysates (WPH) played an important role in inhibiting the oxidation of MP at 4 °C and improving the stability and water-holding capacity of the meat system. Therefore, the aim of this work was to determine the protected effect of WPH on the pork patties and MP undergoing multiple F–T cycles. The effect of WPH content on the lipid and protein oxidation, MP structure and water retention of pork patties was investigated.

## 2. Materials and Methods

### 2.1. Chemicals and Materials

Whey protein isolate (WPI, 95%) or native whey protein (NWP) was obtained from Davisco Foods International, Inc., (Minnesota, MN, USA). Fresh pork longissimus and back fat were collected from a commercial meat processing plant (Yantai, Shandong, China). Alcalase 2.4 L (6 × 10^4^ U g^−1^) was purchased from Novo Nordisk Biochem Inc. (Franklinton, NC, USA). Phosphate buffer, bromophenol blue (BPB), 5,5′-Dithiobis-(2-nitrobenzoic acid) (DTNB), ethylene diamine tetraacetic acid (EDTA), butylated hydroxyanisole (BHA) were obtained from Sigma Chemical Co., Ltd. (St. Louis, MO, USA). All of the other testing chemicals were of analytical grade and purchased from Sinopharm Group Chemical Reagent Co., Ltd. (Shanghai, China).

### 2.2. Preparation of WPH

NWP was configured into a 5% solution, preheated at 95 °C for 5 min, and then transferred to a water bath at 65 °C; 2% Alcalase (6 × 10^4^ U/g) basic protease was mixed. During the enzymatic hydrolysis, 1 mol/L NaOH was added to the solution to maintain the pH at 8.5. Finally, the enzyme was inactivated by a boiling water bath for 5 min after hydrolysis for 5 h. The degree of hydrolysis of whey protein hydrolysate was determined according to the method of Peng et al. [24], and the final degree of hydrolysis reached 35~36%. The whey protein hydrolysate was lyophilized for subsequent experiments.

### 2.3. Preparation of Pork Patties 

The pork longissimus was trimmed of fat and connective tissues. Furthermore, the pork longissimus and back fat at a ratio of 4:1 were minced by a meat grinder to make chopped pork and then randomly divided into 6 groups. The chopped pork sample without any additive was used as the control group, and the other 5 groups were, respectively, added with 10% NWP, 5%, 10% and 15% WPH and 0.02% BHA. Afterwards, 1.5% NaCl was added to the pork mixture for each group. After thoroughly mixing, the meat patties (about 75 g each) were produced using a round mold (diameter: 6.5 cm, thickness: 1.5 cm). The temperature was maintained at around 4 °C during the entire meat patties manufacturing. Finally, polyethylene bags were used to individually wrap the meat patties. All samples were stored at −18 °C for 5 days and then thawed at 4 °C for 12 h until the center temperature reached 0~2 °C, representing one freeze–thaw cycle. According to the above operations, 3, 5, 7 and 9 F–T cycles were repeated. 

### 2.4. Lipid Peroxide Value

The peroxide value (POV) of meat patties was evaluated following a method described by Wang et al. [25]. A 5.0 g minced sample was used to measure POV. The results were expressed as μg/kg of meat.

### 2.5. Extraction of Myofibrillar Protein (MP)

MP was extracted using the method described by Xue et al. [26] with some modifications. The minced pork patty was homogenized in the isolation buffer (0.1 mol/L NaCl, 1 mmol/L EDTA, 2 mmol/L MgCl_2_, 10 mmol/L K_2_HPO_4_, pH 7.0) and centrifugation (3500× *g*, 15 min). The supernatant was discarded, and the pellet was extracted twice more with the isolation buffer as indicated above. The pellet was then washed twice with 0.1 mol/L NaCl. After filtering through four layers of cheesecloth, the adjusted pH to 6.0. MP concentration was determined by the Biuret method. Subsequently, the MP was kept at 4 °C and used within 48 h.

### 2.6. Total Sulfhydryl (SH) Content

The SH content of MP was evaluated using the method according to Yu et al. [27] with minor modifications. An aliquot (1 mL) of MP solutions (4 mg/mL) was supplemented with 9 mL phosphate buffer (8 mol/L urea, 2% sodium dodecyl sulphate and 10 mmol/L ethylenediaminetetraacetic acid, pH 7.0). Then, 5 mL resultant mixture and 1 mL 0.1% DTNB-Tris-HCl buffer (pH 7.0) were mixed and incubated at 40 °C for 30 min. The absorbance was measured at 412 nm. The SH content was calculated using the molar extinction coefficient of 13,600 L·mol^−1^·cm^−1^ and expressed as μmol/g of protein.

### 2.7. Surface Hydrophobicity

The surface hydrophobicity of MP was determined according to the method performed by Yu et al. [27]. In brief, the MP solution was adjusted to 5 mg/mL with phosphate buffer (pH 6.0); 200 μL of 1 mg/mL BPB was added to 1 mL of the MP solution. The same treatment without MP was used as the control. The samples were centrifuged at 2000× *g* for 15 min at 4 °C. Subsequently, the supernatants were separated and diluted 10 times, and their absorbance was measured at 595 nm. The BPB bound content was calculated by the following equation:$$\mathrm{BPB}\;\mathrm{bound}\;(\mu g)=200\;\mu g\;\times \;\frac{A_{\mathrm{control}}-A_{\mathrm{sample}}}{A_{\mathrm{control}}}$$
where 200 μg is the mass of bromophenol blue.

### 2.8. Zeta Potential

The zeta potential of MP was determined using a Malvern Zetasizer Nano ZS90 (Malvern Instruments Ltd, Worcester, Worcester shire, UK) [28]. The MP solution was diluted to 2 mg/mL in a 10 mmol/L phosphate buffer (pH 7.0) and vibrated for 2 h to ensure the samples were homogeneous. After vibration, the MP solution was stirred at 4 °C until required for use.

### 2.9. Particle Size

The particle size and particle size distributions (PSD) of MP solutions were determined by a Mastersizer laser light scattering analyzer (Mastersizer 2000, Malvern Instruments Ltd., Worcester, Worcester shire, UK) in accordance with Zhao et al. [28]. The refractive index was installed to 1.52 for the solution particles and 1.33 for the deionized water. Meanwhile, the absorption coefficient of the dispersed phase was set to 0.01, and the particles were set to non-spherical. The particle size was expressed as volume-weighted (*d*_4,3_) mean diameter and particle size distribution.

### 2.10. Dynamic Rheological Properties

The dynamic rheological properties of meat patties were measured using a Rheometer (MCR301, Anton Paar, Austria) with oscillatory mode according to Zhao et al. [28]; 5.0 g samples were placed between two 50 mm diameter parallel plates (0.4 mm plate gap). Samples were heated from 20 to 80 °C with a temperature control rheometer at a scan rate of 1 °C/min. The tests were conducted at a maximum strain of 0.012 and a frequency of 1 Hz. The storage modulus (G′) was recorded and analyzed for determining rheological behavior.

### 2.11. Pressure Loss

The pressure loss was determined from the description of Farouk et al. [29]. Thawing samples were held under pressure of 35 kg for 5 min, and visible exudates were wiped. The pressure loss was calculated using the following equation:$$\mathrm{Pressure}\;\mathrm{loss}\;(\%)=\;\frac{W_{0}-W_{1}}{W_{0}}\;\times \;100$$
where *W*_0_ and *W*_1_ are the weights of the patties before and after pressure, respectively.

### 2.12. Low-Field Nuclear Magnetic Resonance (NMR) Analysis

The NMR relaxation measurements were performed based on the method of Han et al. [30]. Briefly, approximately 2.0 g of minced samples was put inside the NMR tubes (diameter: 15 mm) and connected with an NMR probe. The measurements of the transverse relaxation time (*T*_2_) were carried out on a Niumag Benchtop Pulsed NMR analyzer (PQ001; Niumag Corporation, Shanghai, China) with a magnetic field strength of 0.5 ± 0.08 T, operating at a frequency of 22.6 MHz. The relaxation times were measured using the Carr–Purcell–Meiboom–Gill (CPMG) sequence, and three relaxation times (*T*_2b_, *T*_21_ and *T*_22_) were recorded as outputs.

### 2.13. Statistical Analysis

The effect of the WPH on the lipid and MP oxidative, MP aggregation and the water-holding capacity of pork patties was assessed using the Statistix 8.1 software package (Analytical Software, St. Paul, MN, USA). Principal component analysis (PCA) was performed between all parameters using the SIMCA software (version 14.1, Umeå, Sweden) to elucidate similarities and differences between samples. Significant differences (*p* < 0.05) were determined by one-way analysis of variance (ANOVA) with Tukey’s multiple comparisons, and the data were expressed as mean ± standard error (SE).

## 3. Results and Discussion

### 3.1. Lipid Peroxide Value

POV is used to assess the extent of lipid oxidation and rancidity, and it can indicate the amount of primary oxidation products (hydroperoxides) formed during lipid oxidation [31]. As shown in Figure 1, the POV of all samples increased gradually with F–T cycles increasing and reached the maximum value of 5.44–6.57 μg/kg after nine F–T cycles. This finding was consistent with Chen et al. [32], who found that the POV of beef increased significantly after the F–T cycles (*p* < 0.05). Lipid oxidation can cause tissue destruction, protein denaturation and muscle fiber damage in pork patties [27]. In general, antioxidants could prevent lipid oxidation and improve the quality of meat products [31]. The POV of the patties with 15% WPH and 0.02% BHA was significantly lower than that of the control sample during each F–T cycle (*p* < 0.05). Especially, the POV of the patties with 15% WPH was significantly decreased by 17.20% compared with that of the control after nine F–T cycles. These results indicated that the addition of 15% WPH exhibited a high antioxidant capacity, which could significantly reduce the formation of lipid free radicals, thus delaying lipid oxidation in the F–T cycles.

### 3.2. Total Sulfhydryl Content

Sulfhydryl is both exposed on the surface and buried within protein molecules, influencing the spatial structure of the protein [33]. Total sulfhydryl content is an important indicator to assess the oxidation and denaturation of proteins. As shown in Figure 2, the total sulfhydryl content of control was 85.73 μmol/g, which reduced with the increase of F–T cycles and reached the lowest value of 36.75 μmol/g after the nine F–T cycles. This result was consistent with the findings of Wu et al. [34], which may be attributed to the fact that sulfhydryl in MP was easily oxidized to disulfide bond during F–T cycles, resulting in a decrease in total sulfhydryl content [5]. The decrease of total sulfhydryl content caused the destruction of protein spatial structure, which may result in protein aggregation and coagulation [34]. The total sulfhydryl content of the patties with WPH was significantly higher than that of the control (*p* < 0.05), and it increased gradually as the WPH concentration increased from 5% to 15% (*p* < 0.05). This may be because WPH overlapped the sulfhydryl groups of actomyosin molecules, contributing to the decrease of oxidation sensitivity of the sulfhydryl group. Similar to our findings, Jonenberg et al. [35] found that natural antioxidants effectively inhibited protein oxidation and increased the total sulfhydryl content of protein. Therefore, when 15% WPH was added to the pork patties, the total sulfhydryl content was significantly higher than that of other samples after F–T cycles. This finding supported the results of POV mentioned above; 15% WPH effectively prevented protein oxidation during F–T cycles and improved the oxidation stability. It should be noted that there was no significant difference in sulfhydryl content between 15% WPH and 0.02% BHA samples during each F–T cycle (*p* > 0.05).

### 3.3. Surface Hydrophobicity

Surface hydrophobicity is related to protein structural denaturation and conformation modification, and it has a significant impact on the physicochemical properties, functional properties and gel stability of MP [36]. As shown in Figure 3, the surface hydrophobicity of all the samples without NWP or WPH increased with the increase of F–T cycles, implying unfolding of MP and exposure of hydrophobic amino acid residues [25]. After F–T cycles, MP molecules were stretched and unfolded, leading to the destruction of hydrophobic and hydrogen bonds in protein molecules, further leading to protein aggregation and functionality loss [37]. Wang et al. [38] demonstrated that oxidation can change the physical properties of protein and increase surface hydrophobicity. Surface hydrophobicity decreased with the addition of WPH during F–T cycles (*p* < 0.05), especially 15% WPH, with no significant difference with 0.02% BHA (*p* > 0.05). The results indicate that the addition of WPH and BHA reduced the extent of denaturation of protein in pork patties, and the protein could maintain its original structure well. The surface hydrophobicity of the sample with 15% WPH was significantly decreased by 30.56% compared with that of the control after nine F–T cycles. These results showed that adding 15% WPH to pork patties could reduce the exposure of nonpolar amino acid side chain groups and the tendency of protein aggregation through intermolecular hydrophobic interactions [39].

### 3.4. Zeta Potential

The zeta potential reflects the characteristics of the electrostatic potential near the particle surface [40]. Higher zeta potential may be due to more charged groups exposed to the protein surface and increased particle repulsion, resulting in increased stability [28]. Proteins with low zeta potentials, on the other hand, tend to coagulate or flocculate. As shown in Figure 4, the zeta potential of MP is negative in all samples, indicating that there are more negatively charged amino acids on the protein surface than positively charged amino acids. As the number of F–T cycles increased, the absolute value of the zeta potential decreased. This result could be attributed to oxidative denaturation of the protein, which leads to the deterioration of the MP stability, and coacervate was formed between the MP after F–T cycles [41].

Increasing surface charge on protein particles may reinforce electrostatic repulsion between particles and prevent further formation of aggregates [42]. Compared with the control sample, the addition of NWP, WPH and BHA significantly increased the zeta potential (*p* < 0.05), especially the 15% WPH, and 0.02% BHA caused the highest zeta potential value. After nine F–T cycles, the absolute zeta potential increased from 12.39 mV to 15.07 mV as WPH concentration increased from 5% to 15%, showing a dose dependency. In particular, the absolute value of the zeta potential with 15% WPH was significantly increased by 146.14% compared with that of the control after nine F–T cycles (*p* < 0.05). Therefore, the addition of 15% WPH could maintain the charge quantity in the MP after F–T cycles, preventing system disorder during protein oxidation through electrostatic interactions and improving protein oxidative stability [43].

### 3.5. Particle Size 

Particle size is a critical indicator of stability in the meat system [44]. The difference in particle size is mainly affected by the denaturation and aggregation of the protein. The enhancement of protein–protein hydrophobic interactions will promote intermolecular cross-linking and the formation of protein aggregates [45]. As shown in Figure 5, the average volume diameter of MP increased during F–T cycles, indicating that F–T cycles could lead to the destabilization of the protein with increased heterogeneity and particle enlargement [46]. This result was consistent with the finding of surface hydrophobicity and the zeta potential (Figure 3 and Figure 4). During F–T cycles, the formation of aggregates may be through intermolecular disulfide bonds, intra-or inter-molecular cross-links and hydrophobic interactions [41]. Furthermore, the MP oxidation would result in more loss of free sulfhydryl groups, the formation of carbonyls and an increase in particle size [16].

Protein with smaller sizes exhibits better emulsification and oxidation stability [47]. The addition of WPH resulted in better stability and smaller particle size in the F–T cycles when compared to the control sample (*p* < 0.05), especially with the 15% WPH sample. Except for the nine F–T cycles, no significant differences in *d*_4,3_ were found between the 15% WPH and 0.02% BHA samples (*p* > 0.05). As a result, the addition of 15% WPH and 0.02% BHA could result in the smallest particle size of protein. The particle size of the sample with 15% WPH was significantly decreased by 34.67% compared with that of the control sample after nine F–T cycles, indicating that WPH decomposed the macromolecular structure of MP into small particles and disrupted the aggregation of proteins.

### 3.6. Rheological Properties

The G′ represents the viscoelasticity behavior of meat protein, and it also describes gel strength. High G′ in meat products indicated improved elasticity, intricacy and gel structure [48]. As shown in Figure 6, changes in G′ revealed that the MP formed a gel in three stages. With the increase of temperature in the first stage (20 °C–55 °C), the G′ of all the samples first showed a slight decrease from 20 °C to 55 °C, mainly because the MP dissolved, swelled and folded during the heating process, resulting in a decrease in G′ [49]; then it gradually increased to the first peak at around 55 °C, which was due to the protein–protein interactions, resulting in gelation [50]. During the second stage (55 °C–60 °C), G′ decreased rapidly and reached a nadir at 60 °C, which may be due to the heat treatment causing the myosin tail to unfold, thus increasing the fluidity of the newly formed gel and destroying the gel structure [51]. During the third stage (60 °C–80 °C), G′ increased rapidly with increasing temperature. The reason is that the increased temperature caused the myosin molecule’s conformation to loosely unfold to expose the active groups, enabling cross-linking to form a firm, irreversible and elastic three-dimensional gel structure [52].

The changed trend of the G′ in different samples was similar during the whole heating process. The results of the control sample show that the freeze–thaw process has a great influence on G′. At 80 °C, the G′ dropped from 15,650 Pa to 11,290 Pa after the nine F–T cycles, indicating that the gel properties of the control MP were damaged severely. The interaction and the formation of intermolecular covalent bonds between proteins were blocked due to F–T cycles [53]. The addition of WPH alleviates the decrease of G′. After nine F–T cycles, the G′ of the patties with NWP, WPH and BHA were higher than that of the control, especially in the samples with 15% WPH and 0.02% BHA. The increase of G′ of the pork patties by adding NWP, WPH and BHA could be related to the increase of sulfhydryl content (Figure 2). The increased sulfhydryl content can reduce rigidity and increase elasticity, resulting in the improvement of rheological properties [53]. Overall, the addition of 15% WPH resulted in more ordered and elastic behavior of the sample, facilitating the formation of three-dimensional protein networks. Therefore, WPH could promote a higher cross-linking between protein molecules, preventing the deterioration of the gel structure caused by oxidation, as well as improving the elasticity and water-holding capacity of the gel network.

### 3.7. Pressure Loss

Pressure loss is a crucial index for determining the water retention capacity and quality of meat products [54]. As shown in Figure 7, the pressure loss in pork patties increased significantly during F–T cycles (*p* < 0.05). Similar results were found by Wang et al. [24], who reported that the pressure loss of F–T cycle samples was significantly higher than fresh samples (*p* < 0.05). The results may be attributed to the mechanical damage caused by recrystallization. In freezing and frozen storage, the ice crystals formed inside or outside the cells and caused a disruption of the cellular membranes. As a result of perimysium leakage, water in the intracellular space moves to the extracellular space, which increases the drip loss of muscle after thawing [55]. In addition, F–T cycles may induce meat oxidation by causing direct texture damage of meat tissues and damage to specific cellular structures, especially membrane lipids, resulting in pressure loss [1]. During F–T cycles, the pressure loss in the samples added WPH and BHA decreased obviously. After seven or nine F–T cycles, the water retention of sample with 15% WPH was significantly higher than that of the sample with 0.02% BHA (*p* < 0.05). The pressure loss of the sample with 15% WPH was significantly decreased by 13.96% compared with that of the control sample after nine F–T cycles. Therefore, adding 15% WPH to pork patties during F–T cycles could effectively improve water retention.

### 3.8. Low-Field Nuclear Magnetic Resonance Analysis

Low-field NMR technique provides information about the mobility and distribution of water in meat systems, especially water-binding capacity to muscle proteins [30]. Longer relaxation time means that the binding ability between water and meat becomes weaker. As shown in Figure 8, three peaks were observed during F–T cycles, which represented that there were three different water phases, including *T*_2b_ (1–10 ms), *T*_21_ (10–100 ms) and *T*_22_ (100–1000 ms). *T*_2b_ with the shortest relaxation time represents the bound water that associated with macromolecules tightly; *T*_21_ represents the immobile water entrapped in the MP network; *T*_22_ with the longest relaxation time represents the free water in the protein lattice [56]. In Figure 8, *T*_2b_ did not change obviously during F–T cycles, indicating that the bound water was tightly bound to the proteins and had high freezing resistance. However, *T*_21_ and *T*_22_ increased during F–T cycles, which is dependent on the change in the spatial structure and a reduction in the water-holding capacity of proteins. The difference in *T*_22_ may indicate that free water molecules in the gel system are dismissed during F–T cycles, increasing the molecular mobility of free water [57].

The changes in *T*_2_ relaxation times were associated with the denaturation of proteins caused by repeated F–T cycles. The results are consistent with Zhang et al. [3], who found that *T*_2b_, *T*_21_ and *T*_22_ in porcine longissimus muscle increased significantly after the F–T cycles, indicating that immobile water was transferred to free water, and free water mobility increased. In general, the mobility of water molecules increased during F–T cycles, thus prolonging the corresponding relaxation time. However, the addition of WPH reduced the *T*_2_; the *T*_2_ of samples with 15% WPH added was shorter than that of the other samples during F–T cycles. The decrease in *T*_2_ may be due to the fact that WPH promotes the formation of protein network structures, thus restricting proton migration related to water and fat molecules [56]. These results indicated that WPH treatment could improve the hydration status of polypeptides by expanding intermolecular spaces and increasing active side chains so that more free water was converted to immobile water, which improved water binding ability and inhibited water mobility, resulting in a tight gel structure and increased elasticity.

### 3.9. Principal Component Analysis

Principal component analysis (PCA) was used to determine the relationship between the oxidation reaction, protein structure and water-holding capacity of pork patties during repeated freeze–thaw cycles. As shown in Figure 9, the first two principal components account for 88.87% of the total variance. Among them, the first principal component (PC1) was the most important variable, accounting for 80.31% of the total variance. The samples after five F–T cycles, seven F–T cycles and nine F–T cycles were located on the positive PC1 axis and positively correlated with surface hydrophobicity, particle size, pressure loss, POV and *T*_2b_, *T*_21_, and *T*_22_ content. In contrast, the samples after no F–T cycles, one F–T cycle and three F–T cycles were located on the negative PC1 axis and positively correlated with the total SH content, G′ and the zeta potential. These results indicated that as the number of repeated freeze–thaw cycles increased, the oxidation degree increased, the protein structure changed, and the water-holding capacity of pork patties decreased. Under the same freeze–thaw cycles, the distribution of samples with 15% WPH was the closest to the distribution of samples without freeze–thaw cycles, indicating that the addition of 15% WPH was more effective at preserving the quality of freeze–thawed pork patties. After nine F–T cycles, the POV, surface hydrophobicity, particle size and pressure loss of the samples with 15% WPH were reduced by 17.20%, 30.56%, 34.67% and 13.96%, respectively, and G′ at 80 °C was reduced by 16.16%. The total SH content and the absolute value of the zeta potential increased by 69.62% and 146.14%, respectively. This indicates that adding 15% WPH to the pork patties can effectively inhibit the oxidation reaction and protein structure changes and improve the water-holding capacity of the pork patties during the F–T cycles.

## 4. Conclusions

The results showed that repeated F–T cycles had a significant impact on the quality of pork patties as well as the oxidative damage to proteins. During F–T cycles, significant differences in physical properties, protein oxidation, structures and elasticity were observed in pork patties with different concentrations of WPH added. Pork patties with 15% WPH had better quality, microstructure and oxidation stability in all samples, similar to the sample with 0.02% BHA. The findings revealed that 15% WPH played an important role in inhibiting protein oxidation and improving the functionality and water retention of pork patties. These results serve as a guide for obtaining fresh pork patties during cryopreservation and improve the further application of WPH in the food industry.

## Figures and Tables

**Figure 1 foods-11-02133-f001:**
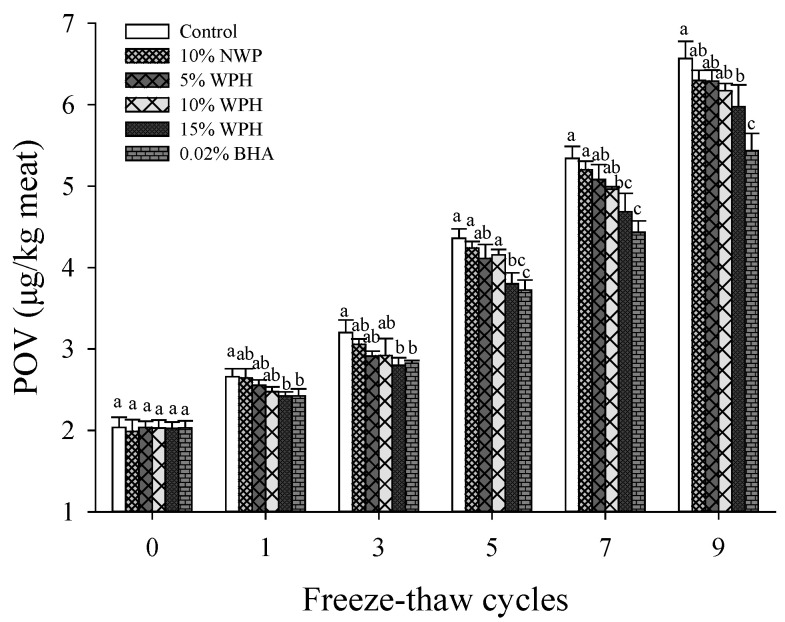
Changes in peroxide value (POV) of multiple freeze–thaw (F–T) cycles of pork patties with different whey protein hydrolysate (WPH) contents. The significant differences among different samples are indicated by different lowercase letters (a–c). Control: without any additives in the sample; NWP: native whey protein; BHA: butylated hydroxyanisole.

**Figure 2 foods-11-02133-f002:**
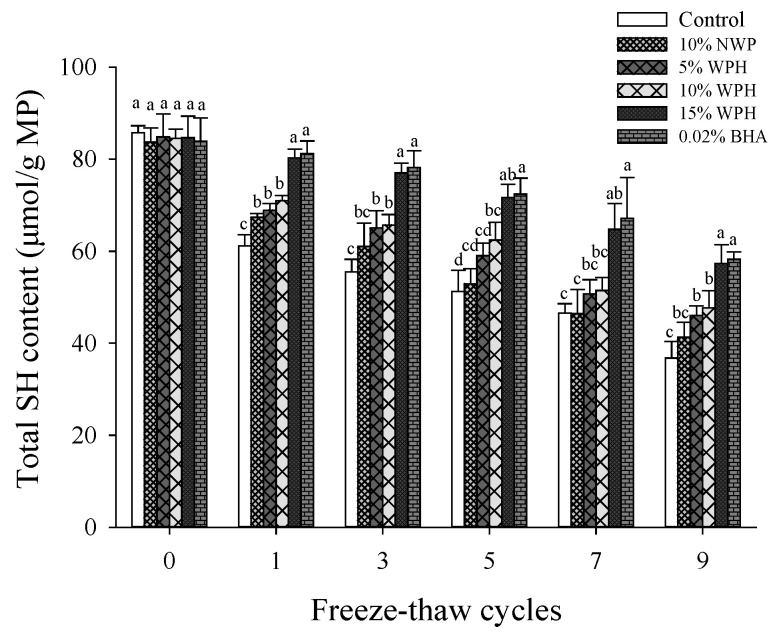
Changes in total sulfhydryl content (SH) of myofibrillar protein (MP) of multiple freeze–thaw (F–T) cycles of pork patties with different whey protein hydrolysate (WPH) contents. The significant differences among different samples are indicated by different lowercase letters (a–d). Control: without any additives in the sample; NWP: native whey protein; BHA: butylated hydroxyanisole.

**Figure 3 foods-11-02133-f003:**
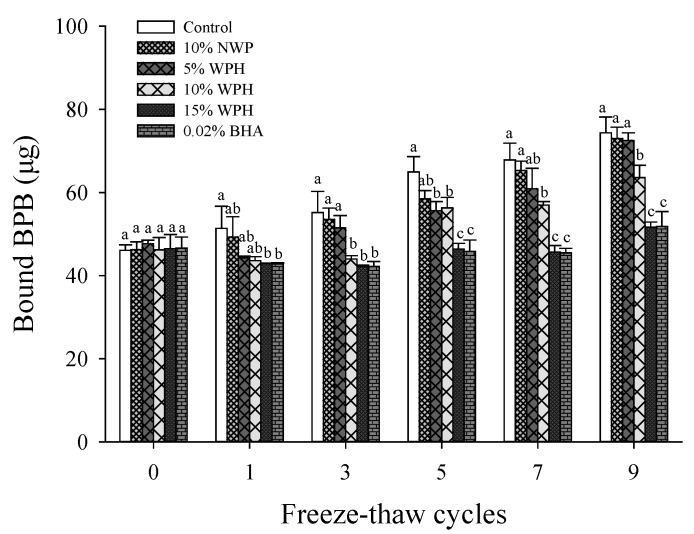
Changes in surface hydrophobicity of myofibrillar protein (MP) of multiple freeze–thaw (F–T) cycles of pork patties with different whey protein hydrolysate (WPH) contents. The significant differences among different samples are indicated by different lowercase letters (a–c). Control: without any additives in the sample; NWP: native whey protein; BHA: butylated hydroxyanisole; BPB: bromophenol blue.

**Figure 4 foods-11-02133-f004:**
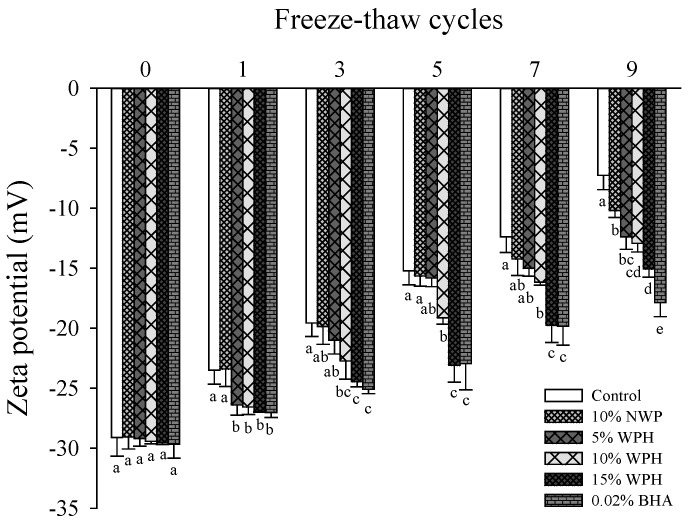
Changes in the zeta potential of myofibrillar protein (MP) of multiple freeze–thaw (F–T) cycles of pork patties with different whey protein hydrolysate (WPH) contents. The significant differences among different samples are indicated by different lowercase letters (a–e). Control: without any additives in the sample; NWP: native whey protein; BHA: butylated hydroxyanisole.

**Figure 5 foods-11-02133-f005:**
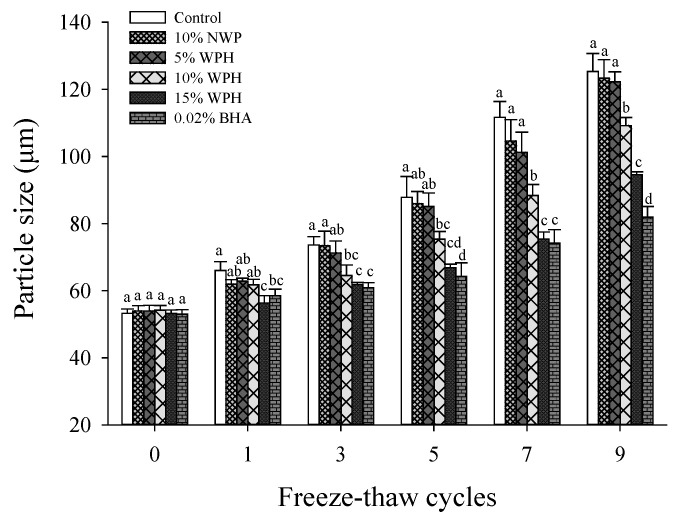
Changes in particle size of myofibrillar protein (MP) of multiple freeze–thaw (F–T) cycles of pork patties with different whey protein hydrolysate (WPH) contents. The significant differences among different samples are indicated by different lowercase letters (a–d). Control: without any additives in the sample; NWP: native whey protein; BHA: butylated hydroxyanisole.

**Figure 6 foods-11-02133-f006:**
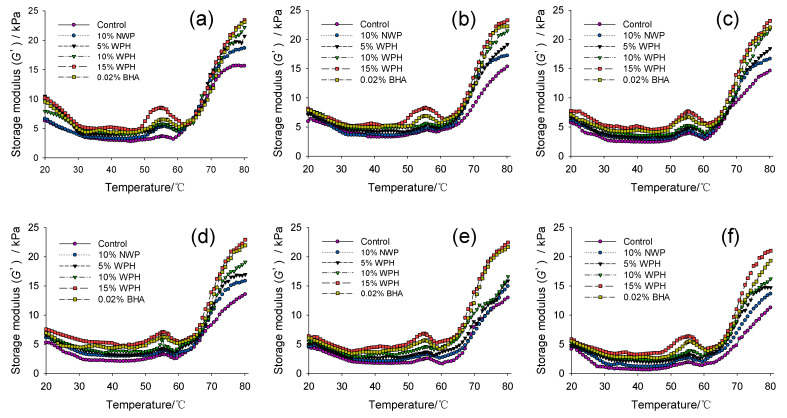
Changes in dynamic storage modulus (G′) of multiple freeze–thaw (F–T) cycles of pork patties with different whey protein hydrolysate (WPH) contents. (**a**–**f**): 0, 1, 3, 5, 7, 9 freeze–thaw cycles. Control: without any additives in the sample; NWP: native whey protein; BHA: butylated hydroxyanisole.

**Figure 7 foods-11-02133-f007:**
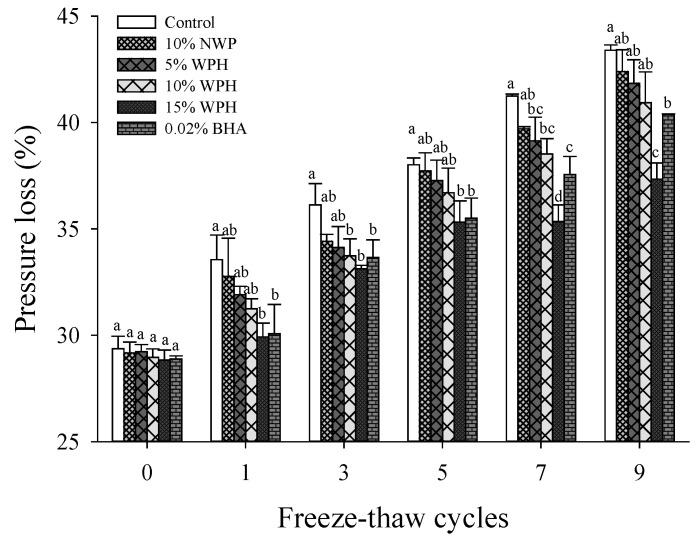
Changes in pressure loss of multiple freeze–thaw (F–T) cycles of pork patties with different whey protein hydrolysate (WPH) contents. The significant differences among different samples are indicated by different lowercase letters (a–d). Control: without any additives in the sample; NWP: native whey protein; BHA: butylated hydroxyanisole.

**Figure 8 foods-11-02133-f008:**
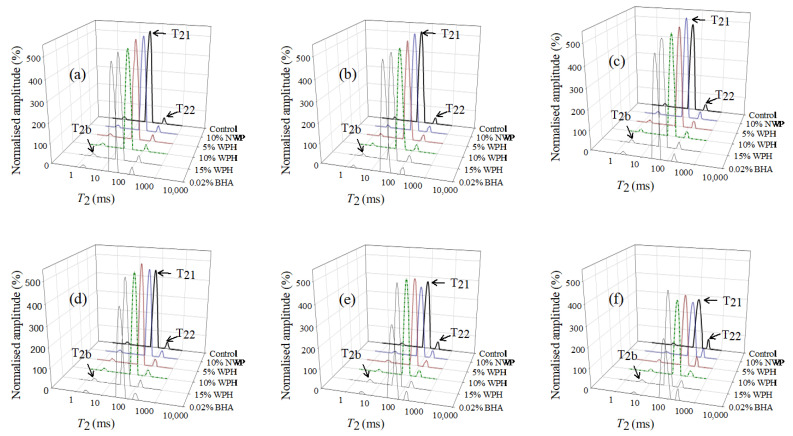
Distribution of the low-field NMR *T*_2_ relaxation times of multiple freeze–thaw (F–T) cycles of pork patties with different whey protein hydrolysates (WPH) contents. (**a**–**f**): 0, 1, 3, 5, 7, 9 F–T cycles. Control: without any additives in the sample; NWP: native whey protein; BHA: butylated hydroxyanisole. *T*_2b_: bound water; *T*_21_: immobilized water; *T*_22_: free water.

**Figure 9 foods-11-02133-f009:**
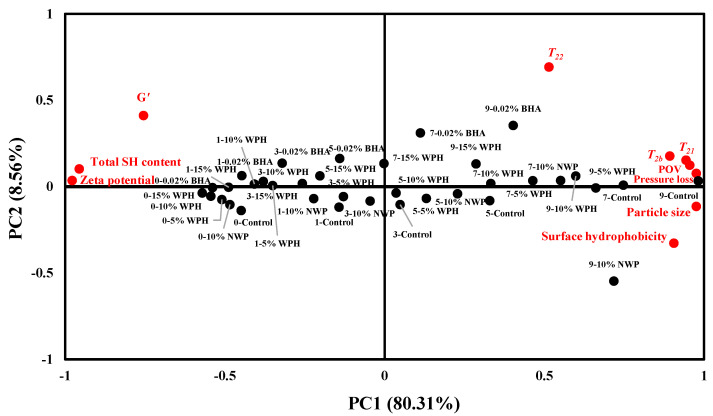
Principal component analysis (PCA) of the oxidative reaction, protein structure and water-holding of multiple freeze–thaw (F–T) cycles of pork patties. PC1: the first principal component; PC2: the second principal component; control: without any additives in the sample; NWP: native whey protein; BHA: butylated hydroxyanisole. *T*_2b_: bound water; *T*_21_: immobilized water; *T*_22_*:* free water; G′: storage modulus; POV: peroxide value; total SH content: total sulfhydryl content.

## Data Availability

The data presented in this study are available within the article.

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
