# Peer review of "Whey Protein Hydrolysates Improved the Oxidative Stability and Water-Holding Capacity of Pork Patties by Reducing Protein Aggregation during Repeated Freeze–Thaw Cycles"

_foods, 2022, doi:10.3390/foods11142133_

Round 1

Reviewer 1 Report

This manuscript is focus on the oxidative stability and water holding capacity by reducing protein aggregation during F-T cycles. The total sulfhydryl content and zeta potential of MP decreased, while peroxide value, surface hydrophobicity, particle size, pressure loss and transverse relaxation times increased. The results showed that adding 15% WPH to pork patties was an effective method to inhibit lipid and protein oxidation, reducing protein aggregation, and improving the water-holding capacity of pork patties during F-T cycles.

Some questions here need to be interpreted:

1. How to control the degree of hydrolysis of Whey protein hydrolysate? What is the degree of hydrolysis?2

2. why BHA adding was selected to be compared?

3. after extraction of MP, does structure of MP change? How to make sure the changes were not induced by extraction process.

4. the discussion of relationship between different indicators were suggested to discuss together.

Author Response

We wish to thank you for the excellent comments and suggestions. We have revised this manuscript according to comments and suggestions. Please see the attachment. We believe the manuscript has been improved, and hope you will find it acceptable.

Reviewer 2 Report

Liu et al. have investigated the effects of incorporating whey protein hydrolysates on improving the oxidative stability and water-holding capacity of pork patties. It is an interesting piece of work and should be of interest to scholars and food chemists in the field. The work is conceptualized and presented well. However, the following points need to addressed for a possible publication of this article in Foods.

1.     Line 23 – “pork patties was an effective method” should be “pork patties can be an effective method”.

2.     Line 100 – specify the inactivation temperature.

3.     Section 2.5 – present the full form of MP with abbreviated form in parenthesis.

4.     L145 – indicate what does “200 mg” represent.

5.     Figure 1-8 – provide the full form of all the abbreviations in the respective captions as each figure should be a standalone that readers should understand the figure contents without referring to the text. For example, POV, NWP, WPH, BHA, MP, SH, F-T, BPB, G¢, LF-NMR, T21, T22, T2b etc.

6.     At the end of section 3, the authors should include a schematic illustration of various parameters studied with their trends in the presence of 15% BHA. This would enable the readers have quick take home trends of all the parameters studied.

7.     It will be highly informative if the authors could run principal component analysis (PCA) on the results of different parameters shown to elucidate the inter-relationships between these parameters.

Author Response

(The authors gave the same response as above.)
